# Use of pyriproxyfen in control of *Aedes* mosquitoes: A systematic review

**John Christian Hustedt**[1,2]*, **Ross Boyce**[3], **John Bradley**[1], **Jeffrey Hii**[2], **Neal Alexander**[1]

**1** MRC Tropical Epidemiology Group, Department of Infectious Disease Epidemiology, London School of Hygiene and Tropical Medicine, London, United Kingdom, **2** Epidemiology Department, Malaria Consortium, London, United Kingdom, **3** Department of Medicine, University of North Carolina at Chapel Hill, Chapel Hill, North Carolina, United States of America

* johnhustedt@gmail.com

**Data Availability Statement:** All relevant data are within the manuscript and its Supporting Information files.

## Abstract

### Background

Dengue is the most rapidly spreading arboviral disease in the world. The current lack of fully protective vaccines and clinical therapeutics creates an urgent need to identify more effective means of controlling *Aedes* mosquitos, principally *Aedes aegypti*, as the main vector of dengue. Pyriproxyfen (PPF) is an increasingly used hormone analogue that prevents juvenile *Aedes* mosquitoes from becoming adults and being incapable of transmitting dengue. The objectives of the review were to (1) Determine the effect of PPF on endpoints including percentage inhibition of emergence to adulthood, larval mortality, and resistance ratios; and (2) Determine the different uses, strengths, and limitations of PPF in control of *Aedes*. A systematic search was applied to Pubmed, EMBASE, Web of Science, LILACS, Global Health, and the Cochrane database of Systematic Reviews. Out of 1,369 records, 90 studies met the inclusion criteria. Nearly all fit in one of the following four categories 1) Efficacy of granules, 2) Auto-dissemination/horizontal transfer, 3) use of ultra-low volume thermal fogging (ULV), thermal fogging (TF), or fumigant technologies, and 4) assessing mosquito resistance. PPF granules had consistently efficacious results of 90–100% inhibition of emergence for up to 90 days. The evidence is less robust but promising regarding PPF dust for auto-dissemination and the use of PPF in ULV, TF and fumigants. Several studies also found that while mosquito populations were still susceptible to PPF, the lethal concentrations increased among temephos-resistant mosquitoes compared to reference strains. The evidence is strong that PPF does increase immature mortality and adult inhibition in settings represented in the included studies, however future research should focus on areas where there is less evidence (e.g. auto-dissemination, sprays) and new use cases for PPF. A better understanding of the biological mechanisms of cross-resistance between PPF, temephos, and other insecticides will allow control programs to make better informed decisions.

**Funding:** Neal Alexander and John Bradley are supported by the UK Medical Research Council (MRC) and the UK Department for International Development (DFID) under the MRC/DFID Concordat agreement, which is also part of the EDCTP2 programme supported by the European Union (Grant Ref: MR/R010161/1). Ross Boyce receives support from the National Institutes of Allergy and Infectious Diseases (K23AI141764) (https://www.niaid.nih.gov/). The funders had no role in study design, data collection and analysis, decision to publish, or preparation of the manuscript.

**Competing interests:** I have read the journal's policy and the authors of this manuscript have the following competing interests: RB is also the PI on an epi study of dengue among children in western Uganda that is funded by Takeda, the maker of one of the tetravalent dengue vaccines. NA is working on the development of a field trial of Integrated Vector Management, including pyriproxyfen, against dengue.

## Author summary

Many important diseases are spread by *Aedes* mosquitoes including dengue, chikungunya, Zika, and yellow fever. Dengue cases are increasing worldwide and there is a lack of effective vaccines and therapeutics. Additionally, mosquitoes have become resistant to commonly used insecticides. Pyriproxyfen (PPF) is an insecticide that prevents juvenile *Aedes* mosquitoes from becoming adults. The objective of this review was to determine the effect of PFF on percentage inhibition of emergence to adulthood, larval mortality, and resistance ratios and determine different use cases, strengths, and limitations. A systematic search was applied to scholarly databases where 67 full text articles met the inclusion criteria. Nearly all included studies fit in four categories, 1) granules, 2) auto-dissemination, 3) ultra-low volume spray, thermal fogging, and fumigant formulations, and 4) mosquito resistance. While mosquito populations were still susceptible to PPF, the concentrations needed to kill a majority of mosquitoes increased among those resistant to temephos (a commonly used insecticide). The evidence is strong that PPF granules do increase immature mortality and adult inhibition, however evidence for other forms and uses is still weak or could be increased. Better understanding of the cross-resistance between PPF, temephos, and other insecticides will allow control teams to make better informed decisions.

## Introduction

Dengue is transmitted through bites of infected *Aedes* mosquitoes, principally *Aedes aegypti* [1]. Although dengue virus infection in humans is clinically apparent in only approximately 25% of cases, it can lead to wide range of clinical manifestations from mild fever to potentially fatal shock syndrome [1]. Despite current research devoted to drug discovery and supportive treatments there is currently no effective antiviral cure for dengue and therefore treatment remains supportive [2]. Dengue infections are caused by four closely related viruses named DEN-1, DEN-2, DEN-3, and DEN-4. Research suggests lifelong immunity is developed after infection but is type-specific [3]. The occurrence of severe symptoms is frequently associated with a secondary infection of a different serotype [1].

Approximately 3.9 billion people in 128 countries are at risk of dengue infection [4]. The disease affects most of the world's tropical and sub-tropical regions and has become the most rapidly spreading mosquito-borne viral disease [1,5]. There were an estimated 390 million infections in 2010, of which 96 million were clinically apparent [4]. These estimates are based on data from various sources including published literature, surveillance data, news reports, and consultations with experts [5]. As the data themselves are of varying quality and completeness the estimates have large confidence intervals. However, the estimates do represent a global consensus of experts that suggests the number of infections is increasing over time and expanding geographically [4]. Between 2010 and 2020 the World health Organization (WHO) is aiming to reduce morbidity and mortality from dengue by least 50% and 25%, respectively [6].

Academics and leading dengue control experts have expressed that regardless of the efficacy of future vaccines, there is growing consensus that no single intervention will be sufficient to control dengue disease [7]. Vector control therefore remains a key part of any dengue control program, and the integration of locally accepted and effective methods is needed [8].These methods together with the development of new vaccines [9], genetic control of mosquitoes [10,11], and new therapeutic drugs [2] will be essential in reducing dengue incidence. In

addition to reducing dengue these vector control methods can also reduce other diseases transmitted by Aedes mosquitoes, principally *Aedes aegypti*, such as chikungunya, Zika, and yellow fever [12]. One insecticide that has been increasingly used is pyriproxyfen (PPF). PPF is a hormone analogue that interferes with the metamorphosis of juvenile *Aedes* mosquitoes, preventing their development into adults capable of transmitting the dengue virus [13]. A recent systematic review assessing the community effectiveness of PPF found it is highly effective in controlling the immature stages of *Aedes* mosquitoes, and to a smaller degree adult *Aedes* populations [14]. Therefore, in this review we are extending the breadth of the systematic review by identifying all evidence (including lab and semi-field studies and use in combined or novel products) on the effect of PPF on *Aedes* mosquitoes.

## Objectives

The objectives of the review were to (1) Determine the effect of PPF on a range of endpoints including percentage inhibition of emergence, larval mortality, and resistance levels (Table 1); and (2) Determine the different uses, strengths, and limitations of PPF in vector control of *Aedes*.

## Methods

### Search strategy and eligibility criteria

This review follows the guidelines as laid out in the Preferred Reporting Items for Systematic Reviews and Meta-Analysis (PRISMA) statement [15] (S1 Table). It was carried out between July 2016 and October 2016, with an update in March-April 2019. All data were extracted by two independent researchers, and discrepancies were resolved by consensus. All studies reporting original data on the use of pyriproxyfen in control of *Aedes or Stegomyia* as a single agent or combined with other control measures were eligible for inclusion. Any study not meeting the inclusion criteria above was excluded.

### Data sources and search strategy

Studies were identified by searching electronic databases, scanning reference lists of articles and consultation with experts in the field. No limits were applied for language in case there was an available English translation. If no translation was available only English and Spanish articles were evaluated. The search was applied to Pubmed, EMBASE, Web of Science, LILACS, Global Health, and the Cochrane Database of Systematic Reviews.

**Table 1. Table of outcome measures in included studies.**

| Outcome measures in included studies | Definition |
|---|---|
| % Larval Mortality | The number of larvae that die/number of larvae included in the experiment |
| % Inhibition of Emergence | The number of mosquitoes that emerge as adults/number of mosquitoes included in the experiment |
| Egg Density Index | The number of eggs/numbers of positive traps |
| Breteau Index | The number of positive containers per 100 houses inspected. |
| Pupal Mortality | The number of pupae that die/The number of pupae included in the experiment |
| Post treatment adult rate ratio | The number of adults collected before the treatment/the number of adults collected after the experiment |
| adult mosquito abundance | The number of adult mosquitos per unit (e.g. house, premise) |

The International Commission on Zoological Nomenclature (ICZN) which governs the nomenclature aspects of zoological taxonomy changed the name of the genus *Aedes* to *Stegomyia* [16]. However, here we follow the suggestion of the *American Journal of Tropical Medicine and Hygiene*, made in consultation with several other journals [17], to continue to use *Aedes* as the genus name. Nevertheless, we have also searched based on *Stegomyia*. The search terms in Table 2 were applied to all databases.

### Study selection

For each search, titles and abstracts were imported into Endnote (Thompson Reuters, Philadelphia, PA, USA), duplicates were removed, and the remaining records were screened. Full texts of potentially relevant records were retrieved and assessed for eligibility, contacting the author of the report as necessary. Reference lists of all potentially eligible articles and reviews were also searched.

A data extraction sheet was developed, and pilot tested on a random selection of included studies and refined accordingly. As many of the studies were not directly comparable (e.g. due to different concentrations, formulations, or combinations of insecticides) a meta-analysis was not attempted. The review protocol was registered with the International Prospective Register of Systematic Reviews (CRD42016046772). The data extraction sheet was used to enter relevant information on each of the included studies, and the resulting tables were used to analyse the effect of PPF. Additional relevant information from each of the studies was entered in a comment section of the extraction sheet for further analysis when writing the manuscript.

## Results

### Search results

The search results are illustrated in S1 Fig. Initially 1,409 records were identified through database searches and 17 additional records were identified through other sources. After screening of title and abstracts, the remaining 109 papers were assessed and reviewed in full, after which 18 articles were excluded. "The most common reasons for exclusion among the full text articles were that papers on PFF reported non-original data, e.g. reviews or perspective pieces (n = 10), or PPF was only tangentially mentioned (again with no relevant original data, n = 8)." A total of 91 studies were then included in the review (S1 Fig).

### Study characteristics

The included studies were published between 1989 and 2018. Six studies were written in Spanish, and the others in English. The studies came from many regions including South America (30%), North America (28%), Asia (17%), Europe (9%), Caribbean (17%), Middle east (5%),

**Table 2. Search Terms Used for Systematic Review.**

| |
|---|
| Pyriproxyfen AND Mosquito Control [MESH] |
| Pyriproxyfen AND Insect Control [MESH] |
| Pyriproxyfen AND Insect Vectors [MESH] |
| Pyriproxyfen AND Disease Vectors [MESH] |
| Pyriproxyfen AND Communicable Disease Control [MESH] |
| Pyriproxyfen AND Dengue |
| Pyriproxyfen AND *Aedes* |
| Pyriproxyfen AND *Stegomyia* |
| (autodissemination of pyriproxyfen) AND *Aedes* |

and Australia (3%). Out of all the 91 studies included, 72 (79.1%) were related to one of the following four core topics and one was related to two of the core topics:

- Efficacy of PPF granules (30 studies) [18–47];

- Auto-dissemination or horizontal transfer of PPF (19 studies) [39,48–66];

- Use of PPF ultra low volume, thermal fogging, and fumigant technologies (15 studies) [67–80];

- Assessing resistance of *Aedes* mosquitoes to PPF (10 studies) [81–90].

Other less common topics were: the use of PPF in novel products (bed nets [65,91], paints [92], release blocks [93], sugar baits [94,95], candles [96], topical treatments [97–99] ovitraps [100,101], resin sticks [102–104], and controlled release mesh [105]; the effect of PPF on the termination of the diapause process [106]; and PPF's environmental persistence and effect on non-target organisms [107]. There was also a review written in 2008 by a PPF manufacturer that focused on the different uses for PPF as a larvicide against nuisance mosquitoes and vectors of dengue and malaria [108].

## Efficacy of PPF granules

PPF granules have been shown to be efficacious in a wide range of lab and field tests in countries across the world. Despite some discrepancies, most studies showed Inhibition of Emergence (IE) near 100% for 90 days at higher concentrations (1–10 parts-per-million (ppm)), and a steady reduction with time post-treatment or with decreasing concentration of active ingredient (Fig 1). Vythilingam et al. found that adult emergence was completely inhibited for four months even with removal and addition of water [32]. However, Richie et al. found that residual PPF detected in water one week later represented just 1.2–1.4% of the total doses applied regardless of the concentration, and the authors highlighted the need to integrate the quick deterioration into any concentration planned for vector management programs [25]. Berti et al. found that increasing the number of larvae treated at 0.05 ppm did not decrease mortality of pupae or adult IE [41].

Studies also suggested that the use of PPF as an alternative to other commonly used insecticides such as temephos or in an integrated method with other means of mosquito control will increase the efficacy with subsequent reduction in the development of resistance [44,83]. Darriet et al. [42] showed the synergetic effect of the rapid killing of mosquito larvae by spinosad (an insecticide based on compounds found in the bacterial species *Saccharopolyspora spinosa*) along with the ability of PPF to kill any pupae that emerged in their trial. Using PPF in combination with other vector control tools (Aquatain AMF or larvicidal oil) was also suggested for emergency control programs [33]. Even using very low doses for short periods has been suggested as a strategy to reduce wild populations before the introduction of genetically modified mosquitoes [25].

## Auto-dissemination or horizontal transfer of PPF

Auto-dissemination or horizontal transfer of PPF is the concept that exploits female mosquitoes to transfer lethal concentrations of an IGR to breeding or resting sites during oviposition, resulting in a reduction of mosquito population by juvenile mortality and subsequent recruitment of adults [48–51,59–66]. The possibility was first tested by forcing adult female and male mosquitoes into contact with PPF coated surfaces in the laboratory [49,60,63]. Studies showed that auto-dissemination occurred, and it successfully increased the mortality rate of larvae that were exposed (See Fig 2).

| First Author | Year | Country | Study Type | Product/Concentration (PPM) | Combination | % Larval Mortality | % Inhibition of Emergence (Concentration in ppm, Time Post-Treatment) | Other |
|---|---|---|---|---|---|---|---|---|
| Al-Azab | 2013 | Saudi Arabia | Lab | Sumilarv® 0.5G (NA) | Diflubenzuron | 5-18% | - | |
| Al-Ghamdi | 2008 | Saudi Arabia | Lab | Sumilarv® 0.5G (0.001, 0.01) | Baycidal (IGR) | - | 20% (0.001), 80% (0.01) | |
| Al-Solami | 2014 | Saudi Arabia | Lab | Sumilarv 0.5G (0.002, 0.02) | None | 10-24% | 24.7(0.002) - 89.2 (0.02) | |
| Ali | 1995 | USA | Lab | TG 97% PPF (NA) | None | 50% (0.00011 ppm); 90% (0.000376) | - | |
| Berti | 2013 | Venezuela | Lab | Sumilarv® 0.5G (000.2, 0.01) | None | - | 77% (N/A, 8 weeks) | |
| Da Silva | 2018 | Brazil | Field | Sumilarv® 0.5G (0.01) | Grass Infusion | - | - | 70 Egg Denstity Index, 120 in Control |
| Darriet | 2006 | France | Lab | TG 98.6% PPF (NA) | Spinosad | 90% (PPF alone) 100% (Combination) | - | |
| De Resende | 2006 | Brazil | Lab | Sumilarv® 0.5G (0.01,0.05) | None | - | 41-98% (0.01, 90 Days) 97.5-100% (0.05, 90 Days) | |
| Kamal | 2010 | Saudi Arabia | Lab | Sumilarv® 0.5G (0.02) | Diflubenzuron | 14% | 91.3% (0.02) | |
| Khan | 2016 | Pakistan | Lab | Sumilarv® 0.5G (0.01-0.05) & Sumilarv® 1.0G (0.01-0.05) | None | 16%-78% | 78% -100% (0.01-0.05) | |
| Lee | 2001 | South Korea | Field | Sumilarv 0.5G® (0.01 - 0.5) | None | - | 61-96% (0.01, 70 days) 100% (0.05, 70 days) | |
| Loh | 1989 | Malaysia | Lab | TG 96.2% PPF (0.00004-0.01) | None | 1.4 - 6.7% | 5.7% (0.00004, 6 hours) 100% (0.01, 6 hours) | |
| Marina | 2018 | Mexico | Lab | Knack CS, 11.2% a.i. (synegenta) | None | 50% (0.020 ppm) | - | 0% positive ovitraps until 7 weeks in dry season (5 weeks in wet season) about 50% by week 12 |
| | | | Field | | | - | - | |
| Mehmood | 2015 | Pakistan | Lab | Predator 0.5® (0.01) | None | - | 100% ( 90 days) - 16% ( 120 Days), 99-100% (45 days) - 1.1% (60 days) | |
| | | | Field | | | - | | |
| Morales | 1997 | Japan | Field | NA (0.1, 1, and 10) | None | - | 47% (0.1), 95.2% (1), 100% (10) | |
| Nayar | 2002 | USA | Lab | Sumilarv 0.5G® (0.02, 0.05) | None | - | 100%/100% (0.2/0.5, 6 Weeks) | |
| | | | Field | | | - | 100%/100% (0.2/0.5, 6 Weeks) | |
| Ocampo | 2014 | Colombia | Field | NA - (50) | None | - | 100% | |
| Ochipinti | 2014 | Venezuela | Lab | Sumilarv® 0.5G (0.01, 0.02, 0.03, 0.04, and 0.05) | None | - | 78.-91.8 (N/A, 90 Days) | |
| Overgaard | 2016 | Colombia | Field | Sumilarv 0.5G® (NA) | None | - | - | Together with deltamethrin treated curtains and jar covers/lids no effect on adult index, but reduction in breteau index compared to control |
| Ritchie | 2013 | Australia | Lab | Sumilarv® 0.5G (0.1, 1, 10, and 100) | None | - | 100% (100, 0-40 days), 100% (10, 0-8 days), 100% (1, 0-4 days), 45% (0.1, 2 days) | |
| Romeo | 2009 | Italy | Field | Sumilarv 0.5G® (NA) | None | - | 70-100% (N/A, 5 weeks) | |
| Sallehudin | 2004 | Malaysia | Lab | Sumilarv® 0.5G (1 and 5) | None | - | 100% (1, 22-28 Days) 100% (5, 36-42 Days) 90% (1, 43-49 Days) 90% (5, 64-70 Days) | |
| Satho | 2002 | Japan | Lab | TG 99% (0.0001, .0001, 0.001) | None | - | Agypti (Tanzania) - 2-3% (0.00001), 17-36% (0.0001), 58-94% (0.001) Albo (Japan1) - 11-27% (0.00001) 52-66% (0.0001), 91-98% (0.001) Albo (Japan2) - 23-30% (0.00001) 41 - 77% (0.0001), 86-88%(0.001) | |
| Seccacini | 2008 | Argentina | Lab | 97.8% PPF - 0.1% sand, 1% surfactant | None | - | 100% (N/A, 45 Days) - 80% (N/A, 180 Days) | |
| Suarez | 2011 | Venezuela | Lab | Sumilarv® 0.5G (0.01, 0.05) | None | - | 66-73.5% (0.01, 4 Weeks) 77-95.7, (0.05, 4 Weeks) | |
| Tuten | 2016 | Switzerland | Lab | 5% I.N.D.I.A. (0.01, 0.5, 2.5, 5) | None | - | 74% (0.01), 83% (0.5), 86% (2.5), 92% (5) | |
| Vythilingam | 2005 | Venezuela | Lab | Sumilarv® 0.5G (0.01, 0.02) | None | - | 100% (0.1, 4 Months), 100% (0.2, 4 Months), 40% (0.2, 6 Months) | |
| Wang | 2013 | Taiwan | Lab | Sumilarv® 0.5G (NA) | None | - | 100% (N/A, 14 Days) | |
| Webb | 2012 | Australia | Lab | Sumilarv® 90CS (NA) | None | - | 100% (10), 20% (1) | |
| Xu | 2010 | China | Lab | Sumilarv® 0.5 (0.06, 0.12) | None | - | 100% | |
| | | | Semi-field | | | - | 99% | |

**Fig 1. Summary of 30 studies included investigating the effect of PPF granules.**

Devine et al. [61] distributed 1-liter plastic pots lined with damp black cloth dusted with pulverized PPF granules in the field and achieved overall reductions in adult emergence of 42–98% thus achieving high coverage of aquatic mosquito habitats. Around the same time, Suman et al. [51], trialed the ability of mosquitoes to auto-disseminate PPF from Ultra-Low Volume (ULV) surface treatments in the field and achieved 15.8% pupae mortality from six weeks in the first year and 1.4% pupae mortality in the second year. ULV spraying is defined as spraying of pesticides at a volume application rate of less than 5 litres/hectare to provide maximum efficacy in killing target vector mosquitoes. Both authors detected that auto-dissemination occurred, however ULV applications were determined not suitable for auto-dissemination. Similar results were found in more recent studies using PPF sprays which found no difference in sentinel containers between intervention and control areas [56]. Better formulations and delivery methods that could load higher doses of PPF and last longer were tested in the lab [59,62] with varying degrees of efficacy.

| Reference | Year | Country | Type of Study | Product/concentration (g/m2) | # of Devices | % Larval Mortality (Time) | % Inhibition of Emergence | Other |
|---|---|---|---|---|---|---|---|---|
| Abad-Franch | 2015 | Brazil | Field | Sumilarv® 0.5G (5) | 100 | 90% | - | |
| Buckner | 2017 | USA | Semi-Field | In2Care | 5 | - | Intervention: 80% (agypti) 90% (albo) ; Control: 20%-30% | |
| Caputo | 2012 | Italy | Lab | Sumilarv® 0.5G (0.5) | 10 | 20% | - | |
| | | Italy | Lab | Sumilarv® 0.5G (5) | 10 | 50-71% | - | |
| Chandel | 2016 | USA | Semi-Field/open | TG PPF (60% a.i.) | 4 | 15% (mean - 8 weeks in 2013)/ 30% (mean 12 weeks 2014) | - | |
| | | | Semi-Field/cryptic | TG PPF (60% a.i.) | 8 | 10% (mean - 8 weeks in 2013)/ 10% (mean 12 weeks 2014) | - | |
| Chism | 2003 | USA | Lab | TG PPF (0.3) | - | 10-30% | - | |
| | | USA | Field | TG PPF (0.4) | - | 50-90% | - | |
| Devine | 2009 | Peru | Field | Sumilarv® 0.5G (5) | 30 | 50% - 80% (Two Sites) | - | |
| Gaugler | 2012 | USA | Lab | TG PPF/NyGuard® (NA) | - | - | Cages-100%; Small Room-80% | |
| Itoh | 1994 | Thailand | Lab | 95.2% TG PPF (1.0) | - | - | 23%-95%; Control 3% | |
| Kartzinel | 2016 | USA | Lab | Esteem® (NA) | 2 | 45% inervention - 1% control | - | |
| | | | Field | Esteem® (NA) | 20 | Site 1 (1%), Site 2 (4-30%, Site 3 (0-12%), Site 4 (0-10%) | - | |
| Llyod | 2017 | USA | Field | Nyguard®- 10% PPF | - | - | - | Overall, there were no differences in pupal mortality between the control and autodissemination vases |
| Mains | 2015 | USA | Field | Esteem® 35 WP IGR/DayGlo® (NA) | - | 70% (Female Experiment) 95% (Male Experiment) | - | |
| Ohba | 2013 | Japan | Semi-Field | TG Sumlarv® 1.0% (w/v) (0.35) | - | Intervention: 50% (20 Days), Control 20% (20 Days) | - | |
| | | Japan | Semi-Field | TG Sumlarv® 0.1% (w/v) (0.035) | - | 50% (6 Days), Control 0% (6 Days) | - | |
| Ponlawat | 2013 | Thailand | Semi-Field | Sumilarv® 0.5G (0.05) | - | - | 25% treated, 10% control | |
| | | | Field | | 4 | - | - | The post-treatment rate ratio (0.4) for treatment area indicated the pyriproxyfen-treated device significantly reduced adult counts during the study period. |
| Sihuincha | 2005 | Peru | Field | Sumilarv® 0.5G (NA) | - | 75% | - | |
| | | Peru | Lab | Sumilarv® 0.5 G - 50, 67, and 83 ppb (direct application) | - | 100% (5 Months) | - | |
| Suman | 2014 | USA | Field | NyGuard® (NA") | - | | - | The sentinel containers for autodissemination showed 15.8% pupal mortality (week 1–6) in the first year, and 1.4% pupal mortality in the second year. No significant difference was detected among the distances and direction for pupal mortality. |
| Suman | 2018 | USA | Field (Essex) -2012 | TG PPF (60% a.i.) - MGK® | 6 | - | - | Pupae Mortality of 15-20% over 12 weeks compared to 3% in Control |
| | | | Field (Hudson) - 2014 | | 24 | - | - | Pupae Mortality of 13.9-20.3% over 8 weeks compared to 1% in Control |
| | | | Field (Hudson) - 2012 | | 1,2,4 | - | - | Pupae Mortality of 10-25% over 12 weeks compared to 5% in Control |
| | | | Field (Middlesex) - 2013 | | 1 | - | - | Pupae Mortality of 50.4% over 8 weeks compared to 5% in Control |
| | | | Field (Mercer) -2012 | | 1 | - | - | Pupae Mortality of 5-10% over 12 weeks compared to 2% in Control |
| Snetselaar | 2014 | Netherlands | Lab | NA | 4 | - | 95%; Control 2% | |
| Tuten | 2016 | Switzerland | Semi-Field | 5% PPF powder - I.N.D.I.A. | 5 | - | - | 3 of 4 trials had staticailly significant difference in pupae between intervention/control |
| Unlu | 2017 | USA | Field | 20% PPF oil & 60% powder | 81 | - | - | Pupal Mortality 12.4% control 0.58% after 50 days, no difference in adults |

**Fig 2. Summary of 19 studies included investigating the effect of auto dissemination or horizontal transfer of PPF.**

Abad-Franch et al. used these auto-dissemination stations in a field trial and found greater than ten-fold rise and greater than a ten-fold decrease in juvenile mosquito mortality and adult mosquito emergence, respectively [48]. However, many of the field studies lacked assays sufficiently sensitive to detect the parts-per-billion concentrations of PPF, therefore limiting the direct evidence of PPF contamination [48]. Although, a recent paper reporting on five different studies done in New Jersey, USA was able to detect PPF by residue analysis in field samples confirming the transfer of PPF by mosquitoes for up to 200 meters [57]. The study used auto-dissemination stations in different contexts and environments and found the stations performed effectively for 8–12 weeks and were able to contaminate 40% of sentinel containers in tire piles 50% in a junkyard. This resulted in the highest pupal mortality in peri-domestic habitats (50%), and sites contaminated with PPF 82%, although the efficacy reduced over time [57].

Snetselaar et al. [50] found 100% IE with the use of a black polyethylene device (In2Care mosquito trap) coated with PPF dust and *Beauveria bassiana* in a semi-field study. *B. bassiana* is a fungus that grows in soils throughout the world and infects mosquitoes reducing their vectoral capacity, inhibiting dengue virus replication, and eventually killing the mosquito [50]. A subsequent semi-field study using the product in Florida found the trap to be attractive to gravid mosquitoes, ability to transfer PPF to sentinel containers, reduce emergence of adult mosquitoes, and reduce survivorship of adult mosquitoes exposed to *B. bassiana* [53].

Main et al. evaluated the use of the "Auto-Dissemination Augmented by Males" (ADAM) approach which used a black plastic device to attract adult females, but also introduces directly treated males (who were exposed to PPF by an insufflator for approximately 5 sec) to transfer PPF to both larval sites and uninfected females during oviposition. Results in a field setting showed 50% reduction in immature mortality rates compared with controls [64]. Field trials demonstrated the ability of PPF-treated males to transmit lethal doses to introduced oviposition containers, both in the presence and absence of indigenous females. A decline in the *Aedes albopictus* population was observed following the introduction of PPF-treated males, which was not observed in two untreated field sites [64]. However, the decrease was not as large as shown by Abad-French et al., which may be due to different environments, susceptibility of the vectors to PPF, different mosquito targets, or different PPF sources and concentrations.

The benefit of auto-dissemination is the potential to effectively counter the main challenge to conventional larviciding approaches by targeting the myriad of cryptic breeding sites that these mosquitoes utilize [48,59]. However, area-wide use requires large amounts of labor when deploying and maintaining numerous stations [63]. Lastly, auto-dissemination efficacy can be affected by several factors such as insecticide resistance, coverage of treated areas, treatment methods, geographical variations and rainfall [51].

## Use of PPF ultra-low volume spray, thermal fogging, and fumigant technologies

Studies investigating the use of PPF in ULV, thermal fogging (TF) and fumigant techniques found IE declining from 100%-50% for 4–6 weeks respectively in treated areas and steadily decreasing with the distance from the sprayer, the length of time from treatment, and type (cold/thermal) of fogging (Fig 3) [67,68,73–79]. Beyond simply having an effect on larval mortality, the sublethal dose of PPF was found to have effects on the fertility and fecundity of adult females who were exposed at the adult stage. Therefore, even if the lethal dose is not achieved, treatment over the long term help decrease the mosquito population through the effects on their reproductive capabilities [76].

Harburguer et al. suggested a strategy including fumigant tablets placed indoors and mixed ULV formulation (including permethrin) for outdoor application [73]. The fumigant they developed showed a high level of recovery of PPF in fumes and resulted in high levels of IE even at low concentrations, as well as an effective knockdown of adults from the permethrin. One limitation of the study data presented above was that they treated only a reduced area (200 houses the same size and evenly spaced and distributed within each of three different treatment areas) and there could have been infestation from adults in nearby households [73].

More recently, studies in Thailand and the USA have shown that multiple spraying machines using combinations of insecticides including PPF were unable to achieve high mortality among *Aedes* mosquitoes placed in hidden (protected) cages, and that the ULV sprays provided better emergence inhibition than the thermal foggers likely due to larger droplet size [70–72].

| Reference | Year | Country | Type of Study | Product/Concentration | Combination | % Larval Mortality | % Inhibition of Emergence (Time-Post Treatment) | comment |
|---|---|---|---|---|---|---|---|---|
| Dantur Juri | 2013 | Argentina | Field | ULV treatment - 3% PPF, Fumigant - 0.2% PPF | Permethrin | 100% | - | |
| Doud | 2014 | USA | Field | Nyguard®- 10% PPF | None | 81.6%-87.4% | 93.13-97.97% | |
| Harburguer | 2011 | Argentina | Field | ULV treatment - 2% PPF, Fumigant - 2% PPF | Permethrin, Methyl 3 | 95.5% | 92.60% | |
| Harburguer | 2011 | Argentina | Field | Fumigant - 2% PPF | Permethrin | 100% | 89.50% | |
| Harburguer | 2012 | Argentina | Field | ULV treatment - 2% PPF, Fumigant - 2% PPF | Permethrin | - | 47-52% Inside, 59.2-71.0% Outside | |
| Harburger | 2014 | Argentina | Lab | 0.2 g/kg PPF | None | - | 20% | |
| | | Argentina | Lab | 2 g/kg PPF | None | - | 40% | |
| Harburger | 2009 | Argentina | Confirmatory | 2 g/kg PPF | Permethrin | - | 95-97% (30 min) | |
| Harwood | 2016 | USA | Field | Nyguard®- 10% PPF | ULB BP-300 | - | - | Sprayers producing larger droplets (misters and cold foggers) were more effective in controlling immature mosquitoes indoors and outdoors. Thermal fogging was more effective in controlling adults indoors, whereas cold fogs and misters were more effective for outdoor control |
| Harwood | 2014 | USA | Semi-Field | Nyguard®- 10% PPF | ULB BP-300 | - | ULV - 50-80% (0-4 Weeks), TF 25-50% (0-4 Weeks) | |
| Lloyd | 2017 | USA | Field | Nyguard®- 10% PPF | None | - | - | The tire pile samples had significantly more mortality (P , 0.0001) out to 4 wk when compared to autodissemination and control vases. |
| Lucia | 2009 | Argentina | Field | 3% PPF | Permethrin | - | Days) | |
| Ponlawat | 2017 | Thailand | Field 1 (Patriot) | Nyguard®- 10% PPF | ULB BP-300 | - | 3.94-21.33 (1 day); -3.35-12.10 (7 days), 1.55-19.78 (14 days) | |
| | | | Field 1 (Twister) | | | - | -4.72-100 (1 day); 1.2-99.29 (7 days), -4.83-97.27 (14 days) | |
| | | | Field 1 (Patriot) | | | - | -9.8-99.57 (1 day); -10.25-68.08 (7 days), -2.99-67.47 (14 days) | |
| Fiorenzano | 2013 | USA | Semi-Field | Nyguard®- 10% PPF | None | - | Direct Treatment -50-100%; Indirect -70%-100% | |
| Unlu | 2018 | USA | Field | Archer IGR - 1.3% PPF | AI lambda-cyhalothrin | - | - | Applications resulted in significant decreases in adult mosquito abundance post-treatment of 74% compared with the untreated control. Both insecticides exceeded the 70% reduction threshold considered as effective for Ae. albopictus control for 2 to 4 wk. However, applications of Archer IGR alone did not reduce adult mosquito abundance. |
| Unlu | 2018 | USA | Field | Nyguard®- 10% PPF | Sumithrin, prallethrin, and *Bti* | - | - | The adult emergence inhibition was significantly higher in the treatment bioassay cups (z=4.65, P<0.0001) and field control bioassay cups (z=8.93, P<0.0001) than controls. They observed a lower trend in adult numbers following the seasonal long combined application of pyriproxyfen and adulticide, with numbers of adult Ae. albopictus in the treatment site up to five times lower than in the control site. |

**Fig 3. Summary of 15 studies included investigating the effect of PPF ultra low volume spray, thermal fogging, and fumigant technologies.**

## Assessing resistance and dose-response relationship of *Aedes* mosquitoes to PPF

Understanding resistance profiles of juvenile and adult mosquitoes is key in public health control programs. Numerous papers reviewed the susceptibility of *Aedes* to PPF and examined cross resistance among PPF and other insecticides (especially temephos). Data show IE levels of 70–100% for 250 days among higher concentrations with levels decreasing with lower concentrations and extended post-treatment time (Fig 4) [38,81–83,85–87,109].

Even among temephos-resistant mosquito populations IE levels show susceptibility to PPF at higher concentrations with the exception of a Florida population already resistant to two other Insect Growth Regulators (IGRs) and dichlorodiphenyltrichloroethane (DDT) [86]. However, their data showed standard larvicides and pyrethroids used for mosquito control were still effective [86]. Indeed, this is opposite of most other studies reviewed here showing resistance of *Aedes* populations to standard larvicides and pyrethroids and susceptibility to IGRs [81–85,87–90].

Rodriguez Coto et al. [110] and Teran Zavala et al. [87] both showed that temephos-resistant strains had similar resistance ratios to reference strains and worked well even at concentrations below World Health Organization (WHO) recommended levels (Resistance Ratio is the measure of resistance in an insect population, calculated by dividing the lethal dose of a study population by the lethal dose of the susceptible population) However, three other studies [83,109,110] found that while mosquito populations were still susceptible to PPF, the lethal concentrations increased among temephos resistant mosquitoes compared to reference strains. Marcombe et al. [86] noted that as PPF has never been used in public health programs

| Reference | Year | Country | Type of Study | Product/Concentration (ppm) | Comparison | Resistance Ratios | % Inhibition of Emergence (Concentration in ppm, Time Post-Treatment) |
|---|---|---|---|---|---|---|---|
| Adrighetti | 2008 | Brazil | Lab/Semi-Field | TG 98.5%/Sumilarv® 0.5G (0.05) | Temefós Fersol 1G | 1.4-6.5 | 100% (N/A, 44 Days) |
| Darriet | 2010 | Martinique | Semi-Field | Sumilarv® 0.5G (0.02) | Spinosad | - | 80% (N/A, 150 days), Combination - 80% (N/A, 250 days) |
| | | Martinique | Field | Sumilarv® 0.5G (0.02) | Spinosad | - | 80% (N/A, 21 days), Combination - 80% (N/A, 126 Days) |
| Gomez | 2011 | Argentina | Lab | TG 97.8% (NA) | Temephos, *Bti*, Permethrin | - | 50% (0.01642-0.00774) |
| Lau | 2015 | Malaysia | Lab | Sumilarv® 0.5G (NA) | None | 1.4 | - |
| Lau | 2018 | Malaysia | Lab | | Methoprene, Difubenzuron, Novaluron, Cyromazine | 0.09 | - |
| Leyva | 2010 | Cuba | Field | 97% PPF (NA) | None | 0.5-3.4 | - |
| Marcombe | 2011 | Martinique | Lab | TG 98.7% (NA) | *Bti*, Temephos, Spinosad, and Diflubenzuron | 2.2 | - |
| | | Martinique | Semi-Field/Field | Sumliarv® 0.5G (0.2, 0.5) | *Bti*, Temephos, Spinosad, and Diflubenzuron | - | Semi-field - 80% (0.05, 250 days), 80% (0.02, 160 days), Field - 80% |
| Marcombe | 2014 | USA | Lab | TG 99.1% (NA) | *Bti*, Temephos, Propoxur, Spinosad, Methoprene | 0.38-2.36 | - |
| Leyva | 2013 | Cuba | Lab | TG 97% (NA) | None | - | 30-40% (1) to 100% (10) |
| Teran Zavala | 2014 | Ecuador | Lab | TG 97% (NA) | Temephos | 4.2-9.2 | Temephos Resistant - 40% (1), 100% (10); Susceptible - 100% (0.1-50) |

**Fig 4. Summary of 10 studies included investigating resistance of *Aedes* mosquitoes to PPF.**

in the United States, it is possible the cross-tolerance of mosquito larvae to IGRs has arisen through the extensive use of temephos for vector control. Significant differences in detoxification enzyme activities in several resistant *Aedes albopictus* populations in the USA suggests the involvement of a metabolic based resistance mechanism [86].

## Safety

PPF has a very favorable mammalian toxicity profile [49]. Even treatment of drinking water at a dosage of 0.01 ppm may be used, which is 30,000 times the lethal dose for mosquitoes and six times the recommended field application rate [32,111,112]. However, as with any chemical there are still concerns regarding environmental impact of the long-term application of PPF in permanent water bodies highlighting the need for environmental studies supporting such uses [13].

## Discussion

The results of this systematic review, which we believe to be the most comprehensive to date, suggest that PPF can effectively control the emergence of adult *Aedes* mosquitoes across a wide variety of environments and in a variety of forms (e.g. granules, ULV sprays, TF, and fumigants). Utilizing a product with a favorable safety profile is especially important in settings where dose recommendations may not always be followed strictly.

Unsurprisingly, the results show the most common use of PPF in granule form results in near 100% (IE) for 90 days at higher concentrations even with removal and addition of water and regardless of the larval density [41]. Integrating PPF with other means of mosquito control (e.g. spinosad) can increase the efficacy with reduction of the risk of resistance development [113,114]. Also, the efficacy of PPF may be altered for populations of mosquitoes known to be resistant to organophosphates (temephos). In areas where the main sources of larval biomass are identifiable and accessible, such as in rural areas with large water storage jars, controlled release PPF granules or matrixes could be quite effective.

Although PPF works well in large water containers, other cryptic or subterranean breeding sites may require significant additional work of control teams to reach. One potential solution is to utilize auto-dissemination or horizontal transfer of PPF. Evidence shows that auto-dissemination occurs, and it successfully increases the mortality rate of larvae that were exposed while reducing the number and viability of eggs from exposed females. Field trials suggest that PPF can increase juvenile mosquito mortality and reduce adult mosquito emergence, however the effect tends to reduce over time, and it is still low enough that additional tools may need to be used in combination with PPF to reduce *Aedes* populations to zero (e.g. granules or controlled release devices for key containers). Significant work has been published on this topic the past two or years illuminating the preferable methods of employment, and the design and spacing of devices. However, there are still no WHO prequalified auto-dissemination devices that can be purchased at large scale for control programs to use, even were they to be recommended. Future studies should look further at defining optimum design of devices and standardized approach for application of PPF dust.

In areas where *Aedes* breeding is located in large outdoor areas where key containers are not present or easy to identify, the use of PPF in ULV, TF and fumigants may be appropriate. Results show IE in treated areas near 100% and steadily decreasing with the distance from the sprayer, the length of time from treatment, and type (cold/thermal) of fogging. Sublethal doses of PPF were also found to have effects on the fertility and fecundity of adult females, suggesting positive effects may reach greater distances away from the sprayer.

Regardless of how effective different PPF products are at distributing the active ingredient, effectiveness can be reduced or lost if the mosquito develops resistance. The results suggest that even among temephos-resistant mosquito populations IE levels show susceptibility to PPF at higher concentrations, with the exception of one Florida population [86]. Many studies found that while mosquito populations were still susceptible to PPF, the lethal concentrations increased among temephos-resistant mosquitoes compared to reference strains. This is even true in areas where PPF has not been used, suggesting the possible cross-tolerance of mosquito larvae to IGRs has arisen through the extensive use of temephos. Therefore, in areas where there is already increased resistance to PPF, control programs should consider combining insecticides that work in synergy. Regular entomological surveillance to monitor the susceptibility status of *Aedes* mosquitoes can help provide evidence and prevent development of resistancex [83].

One of the limitations reported is the issue of compliance by community in areas where top-down government control programs are not distributing PPF [14]. This is due to false perceptions by the community that PPF is ineffective as it mainly acts on late instars and people may continue to observe live early instar larvae [115]. Qualitative studies are required to better understand what communication methods and materials would be most effective to increase community participation in vector control activities.

One of the limitations of this review is the intentionally broad scope and focus on the effect of PPF on *Aedes* rather than the community effectiveness of PPF products on the reduction of dengue. However, a recent review of the community effectiveness of pyriproxyfen as a dengue vector control method found "community participation and acceptance has not consistently been successful and needs to be further assessed. While all studies measured entomological endpoints, only two studies measured the reduction in human dengue cases, with inconclusive results."[14] Additionally, another limitation is the lack of meta-analysis as the heterogeneity of the studies would make the interpretation very difficult and possibly misleading. Hopefully, more field studies will be done using the same concentrations in the future to allow such a meta-analysis.

Future studies can focus on further evaluating new PPF products and new use cases for established products. It will also be important to understand the effectiveness of these products

in Africa. The majority of studies represented here come from Central/South America and Asia, and none from India or Africa. However, global estimates suggest Africa's dengue burden to be equivalent to that of the Americas (16%) and together Africa and India contribute 50% of dengue cases [4]. It will be important to document the effectiveness of these products in these highly endemic areas [14].

In conclusion, the evidence for the effectiveness of PPF to increase *Aedes* larval mortality and IE is strong and consistent. However, the strength of the evidence for different products and use cases varies considerably. PPF granules have highly documented and consistent results that suggest it is very effective especially when used in slightly higher doses and distributed every 30–40 weeks. The use of PPF dust for auto-dissemination and the use of PPF in ULV, TF and fumigants are encouraging although the evidence in favor of them is not as strong or consistent. Many additional novel products have been evaluated (e.g. bed nets, paints, candles, ovitraps), however evidence for these products is very weak at the moment. Future research should focus on these areas where the evidence is less strong and include additional use cases that may become developed in the future. Additional research is also needed to elucidate the biological mechanisms of cross-resistance between PPF, temephos, and other insecticides to allow control teams to make better informed decisions on which products to recommend and procure.

## Supporting information

**S1 Fig. PRIMSA flow diagram.**
(DOCX)

**S1 Table. PRISMA checklist.**
(DOC)

## Author Contributions

**Conceptualization:** John Christian Hustedt, John Bradley, Jeffrey Hii, Neal Alexander.

**Data curation:** John Christian Hustedt.

**Formal analysis:** John Christian Hustedt.

**Methodology:** John Christian Hustedt, Ross Boyce, John Bradley, Jeffrey Hii, Neal Alexander.

**Validation:** Ross Boyce, Neal Alexander.

**Writing – original draft:** John Christian Hustedt.

**Writing – review & editing:** Ross Boyce, John Bradley, Jeffrey Hii, Neal Alexander.

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
