## [Decision Letter · Decision Letter 0]

3 Dec 2019

Dear Mr. Hustedt:

Thank you very much for submitting your manuscript "Use of pyriproxyfen in control of Aedes mosquitoes: a systematic review" (#PNTD-D-19-01379) for review by PLOS Neglected Tropical Diseases. Your manuscript was fully evaluated at the editorial level and by independent peer reviewers. The reviewers appreciated the attention to an important problem, but raised some substantial concerns about the manuscript as it currently stands. These issues must be addressed before we would be willing to consider a revised version of your study. We cannot, of course, promise publication at that time.

We therefore ask you to modify the manuscript according to the review recommendations before we can consider your manuscript for acceptance. Your revisions should address the specific points made by each reviewer. 

When you are ready to resubmit, please be prepared to upload the following:

(1) A letter containing a detailed list of your responses to the review comments and a description of the changes you have made in the manuscript.

(2) Two versions of the manuscript: one with either highlights or tracked changes denoting where the text has been changed (uploaded as a "Revised Article with Changes Highlighted" file); the other a clean version (uploaded as the article file).

(3) If available, a striking still image (a new image if one is available or an existing one from within your manuscript). If your manuscript is accepted for publication, this image may be featured on our website. Images should ideally be high resolution, eye-catching, single panel images; where one is available, please use 'add file' at the time of resubmission and select 'striking image' as the file type. 

Please provide a short caption, including credits, uploaded as a separate "Other" file. If your image is from someone other than yourself, please ensure that the artist has read and agreed to the terms and conditions of the Creative Commons Attribution License at http://journals.plos.org/plosntds/s/content-license (NOTE: we cannot publish copyrighted images). 

(4) If applicable, we encourage you to add a list of accession numbers/ID numbers for genes and proteins mentioned in the text (these should be listed as a paragraph at the end of the manuscript). You can supply accession numbers for any database, so long as the database is publicly accessible and stable. Examples include LocusLink and SwissProt.

(5) To enhance the reproducibility of your results, we recommend that you deposit your laboratory protocols in protocols.io, where a protocol can be assigned its own identifier (DOI) such that it can be cited independently in the future. For instructions see http://journals.plos.org/plosntds/s/submission-guidelines#loc-methods

While revising your submission, please upload your figure files to the Preflight Analysis and Conversion Engine (PACE) digital diagnostic tool, https://pacev2.apexcovantage.com/ PACE helps ensure that figures meet PLOS requirements. To use PACE, you must first register as a user. Then, login and navigate to the UPLOAD tab, where you will find detailed instructions on how to use the tool. If you encounter any issues or have any questions when using PACE, please email us at figures@plos.org.

We hope to receive your revised manuscript by Feb 01 2020 11:59PM. If you anticipate any delay in its return, we ask that you let us know the expected resubmission date by replying to this email.

To submit a revision, go to https://www.editorialmanager.com/pntd/ and log in as an Author. You will see a menu item call Submission Needing Revision. You will find your submission record there. 

Sincerely,

Roberto Barrera, Ph.D.

Associate Editor

Eric Dumonteil

Deputy Editor

Reviewer's Responses to Questions

**Key Review Criteria Required for Acceptance?**

**Methods**

-Are the objectives of the study clearly articulated with a clear testable hypothesis stated?

-Is the study design appropriate to address the stated objectives?

-Is the population clearly described and appropriate for the hypothesis being tested?

-Is the sample size sufficient to ensure adequate power to address the hypothesis being tested?

-Were correct statistical analysis used to support conclusions?

-Are there concerns about ethical or regulatory requirements being met?

Reviewer #1: For the methods section, please highlight better the inclusion criteria and add exclusion criteria.

Please add how the analysis of the massive amount of data from 91 papers was conducted.

Was a quality assessment of the studies done? This needs to be added.

Reviewer #2: Yes

Reviewer #3: (No Response)

**Results**

-Does the analysis presented match the analysis plan?

-Are the results clearly and completely presented?

-Are the figures (Tables, Images) of sufficient quality for clarity?

Reviewer #1: For the descriptive part of the results section, please add numbers for the exclusion of papers. 

An analysis by authors/groups of authors would be useful, please add.

Reviewer #2: Yes

Reviewer #3: (No Response)

**Conclusions**

-Are the conclusions supported by the data presented?

-Are the limitations of analysis clearly described?

-Do the authors discuss how these data can be helpful to advance our understanding of the topic under study?

-Is public health relevance addressed?

Reviewer #1: Good

Reviewer #2: Yes

Reviewer #3: (No Response)

**Editorial and Data Presentation Modifications?**

Reviewer #1: Figures in the tables are corresponding to the references for figures in the text? E.g. figure 3 is referenced in the text in line 182, and figure 2 follows in line 183?

Figure 1,3,4: please add a column about the study design, with the key characteristics. As done in figure 2. 

PRISMA flowchart: please add some boxes to help reading the flowchart

Reviewer #2: Comments and suggestions provided on attached file should improve the manuscript and make it acceptable for publication in PNTD.

Reviewer #3: (No Response)

**Summary and General Comments**

Reviewer #1: This is a remarkably well conducted and written systematic review.

Reviewer #2: See attached comments

Reviewer #3: (No Response)

PLOS authors have the option to publish the peer review history of their article (what does this mean?). If published, this will include your full peer review and any attached files.

Reviewer #1: No

Reviewer #2: No

Reviewer #3: No

---

## [Editor Report · Decision Letter 1]

10 Mar 2020

Dear Mr. Hustedt,

We are pleased to inform you that your manuscript 'Use of pyriproxyfen in control of Aedes mosquitoes: a systematic review' has been provisionally accepted for publication in PLOS Neglected Tropical Diseases.

Best regards,

Roberto Barrera, Ph.D.

Associate Editor

Eric Dumonteil

Deputy Editor

---

## [Editor Report · Acceptance letter]

20 May 2020

Dear Mr. Hustedt,

We are delighted to inform you that your manuscript, "Use of pyriproxyfen in control of *Aedes* mosquitoes: a systematic review," has been formally accepted for publication in PLOS Neglected Tropical Diseases.

Best regards,

Serap Aksoy

Editor-in-Chief

Shaden Kamhawi

Editor-in-Chief
